

# Mid-Late Holocene event registered in organo-siliciclastic-sediments of Lagoa Salgada carbonate system, Southeast Brazil

Anna Paula Soares Cruz[1], Cátia Fernandes Barbosa[1]*, Angélica Maria Blanco[1], Camila Areias de Oliveira[1], Cleverson Guizan Silva[2], José Carlos Sícoli Seoane[3]

[1]Programa de Pós-Graduação em Geoquímica, Departamento de Geoquímica, Universidade Federal Fluminense, Outeiro São João Batista, s/n, Centro, Niterói, Rio de Janeiro CEP 24.020-141, Brazil.
[2]Programa de Pós-Graduação em Dinâmica dos Oceanos e da Terra, Departamento de Geologia, Universidade Federal Fluminense, Niterói, Rio de Janeiro, CEP 24.210-346, Brazil.
[3]Programa de Pós-Graduação em Geologia, Departamento de Geologia, Universidade Federal do Rio de Janeiro, Rio de
Janeiro, CEP 21.941-916, Brazil.

*Correspondence to*: Catia F. Barbosa (catiafb@id.uff.br)

**Abstract.** The formation of Paraiba do Sul river delta plain in the coast of Rio de Janeiro state, Brazil, gave rise to diverse lagoons formed under different sea level regimes and climate variations. Sedimentary core lithology, organic matter geochemistry, and isotopic composition ($\delta^{13}$C and $\delta^{15}$N) were analyzed to interpret the sedimentation of the
paleoenvironment of the Lagoa Salgada carbonate system. Different lithofacies reflect variations of the depositional environment. The abundance of silt and clay between 5.8 to 3.7 ka B.P., enhances the interpretation of a transgressive system, which promoted the stagnation of coarse sediment deposition due to coast drowning. Geochemistry data from this period (5.8-3.7 ka B.P.) suggest the dominance of a wet climate, with an increase of C3 plant and a marked dry event between 4.2-3.8 ka B.P. This dryer event also matches with previous published records from around the world, indicating a
global event at 4.2 ka B.P. Between 3.8-1.5 ka B.P., Lagoa Salgada was isolated, sand and silt arrived at the system by erosion with the retreat of the ocean and less fluvial drainage. Geochemistry from this moment marks the changes to favorable conditions for microorganisms active in the precipitation of carbonates, forming microbial mats and stromatolites in the drier phase.

## 1 Introduction

Severe and prolonged drought around the world characterize the 4.2 ka BP climatic event, and is reflected in proxy records from North America (Booth et al., 2005), Asia (Perşoiu et al., 2018; Scuderi et al., 2019), Africa (Damnati et al., 2012; Gasse, 2000), South America (Tapia et al., 2003), Arabian sea (Giesche et al., 2019), and Antarctica (Staubwasser and Weiss, 2006). This significant aridification event in the mid-late Holocene is recognized in lake sequences, ice cores, and in speleothem, dust and sediment samples. This drought was one of the most pronounced climatic events of the Holocene, after
8.2 ka BP, which was associated with the collapse of several human civilizations in many sites in the world such as in North Africa, the Middle East and Asia (Cullen et al., 2000; Gasse, 2000; Weiss et al., 1993).



The 4.2 event has been the focus of several works. However, the forcing mechanisms behind this event are still unknown. Some authors try to explain the drought and increase in the aridity as a result of the weakening of the Asian monsoon (Kathayat et al., 2018; Wang, 2005) due to the southward migration of the ITCZ. Others suggest a prolonged northward shift of the mean position of the ITCZ (Li et al., 2018), being in contrast with the southward shift of the tropical rain belt. The
irregular fluctuation of atmospheric pressure over the North Atlantic Ocean changing the direction of the cyclonic North Atlantic westerlies (Cullen et al., 2002; Kushnir and Stein, 2010) has been argued as another mechanism that results in mega-drought during 4.2 ka BP, as well as the ENSO conditions linked with drought in the monsoon region contributing to aridity in tropical South America during the same period (Davey et al., 2014).

This dry environmental condition is registered in lake sediment as well as the dynamics and process that occurred in the
water column. Some studies of lake systems have considered the stable isotopes of C and N in sediments as proxies of organic matter (OM) cycles in aquatic systems over time (Salomons and Mook, 1981) and the vegetation changes as a result of the climate alteration (wet/dry condition). Many studies involving OM have been done to characterize past and recent depositional environments (Megens et al., 2002; Pessenda et al., 2004; Salomons and Mook, 1981). The percentage of Total Organic Carbon (% TOC) and the C:N ratio can also indicate the productivity, and OM sources in paleoclimatic
interpretations (Hartmann and Wünnemann, 2009). Thus, the objectives of this work were to evaluate the depositional processes related to sea level changes during the marine and lacustrine stage, and interpret the Holocene climatic changes during the last 5.8 ka BP.

## 2 Material and Methods

Sediment core S-15 was sampled from Lagoa Salgada in Rio de Janeiro State, Brazil (21°54'46.30" S, 41°0'41.70" W),
recovering 212 cm length with a vibracore sampler (Fig.1). Samples were collected every 2 cm for total organic carbon ($C_{org}$) and carbon stable isotopes ($\delta^{13}C_{org}$, $\delta^{15}N$) on bulk organic matter, and every 4 cm for grain size analysis. Sixteen samples throughout the core were analyzed for Fe/Ca. Cores with previously published data and respective authors are mentioned in Fig. 1 and Table1.

### 2.1 Radiocarbon and age model

Radiocarbon analyses were performed at the Arizona Accelerator Mass Spectrometry Facility and BETA Analytic Inc., using 14C accelerator mass spectrometry (AMS). The age model is based in 11 radiocarbon dates from organic material of bulk dried sediment samples and converted to calendar age (Table 2). Radiocarbon dates were calibrated using the R script BACON version 2.227 with IntCal 13 calibration curve to convert to calendar age. The parameters used were mem.mean=0.7, acc.shape=0.8. and t.a=33/t.b=34 (Fig. 2).





### 2.2 Grain size analysis

About 2 g of dried sample were decarbonated using HCl for several hours, centrifuged and washed with distilled water. Hydrogen peroxide ($H_2O$) was also added to remove the organic matter. After these processes, about 30 ml of deflocculant

solution ($Na_{16}O_{43}P_{14}$ – 4 %) was added for 24 hours (Barbosa, 1997). The grain-size measurements were performed using a laser particle analyzer (CILAS 1064), which has a detection range of 0.02–2000 μm, using the grain size statistics method of Folk and Ward (1957) performed in GRADISTAT software version 8.0 (Blott and Pye, 2001).

### 2.3 Total organic carbon, stable carbon and nitrogen isotopes in bulk organic matter

Sediment samples for $C_{org}$, $\delta^{13}Corg$ and $\delta^{15}N$ were dried at 40 °C, powdered and homogenized with an agate mortar. Samples

were decalcified with a 1N HCl solution for several hours, centrifuged and washed with distilled water and subsequently dried at 40 °C. About 30 mg of the dried material was weighed in tin capsules and analyzed at the University of California, Stable Isotope Facility (Davis, USA), using Micro Cube elemental analyzer (Elemental Analyses System GmbH, Hanau, Germany) interfaced to a PDZ Europa 20-20 isotope ratio mass spectrometer (Sercon Ltd., Cheshire, UK). The long-term standard deviation was 0.2 ‰ for $\delta^{13}C_{org}$. The $\delta^{13}C_{org}$ were given as ‰ in relation to Vienna Pee Dee Belemnite (VPDB) and

the $\delta^{15}N$ were given as ‰ in relation to the air.

The carbon accumulation ($C_{org}$ accumulation.) was determined using the following equation of Thunell et al. (1992), Eq. (1):

$$C_{org} \text{ accumulation } (g*cm^{-2}*ka^{-1}) = \rho SR \ (C_{org})  \tag{1}$$

where $\rho$ is density (in $g*cm^{-3}$), SR is the sedimentation rate (in $cm*ka^{-1}$) and $C_{org}$ represents the total organic carbon content.

About 20 mg of the sample were dried, crushed, and placed in specific container to analyze the carbonate content. The analysis was performed every 2 cm resolution using inorganic carbon analyzer (TOC-V with ASI-V SSM 5000 Shimadzu).

### 2.4 Fe/Ca analysis

The Fe/Ca analysis was performed in 16 dried and powdered samples using X-ray fluorescence (XRF) spectrometer Epsilon 3 (PANalytical) at Universidade Federal Fluminense, Brazil.

### 3 Results

The sediment core S-15 recovered the last 5.8 ka BP. The sedimentation rate ranged from 10 to 250 $cm*ka^{-1}$. Sedimentation rate increased between 5 to 4 ka BP (from 10 to 250 $cm*ka^{-1}$) with a posterior decrease during 4 to 3.7 ka BP (from 250 to

140 $cm*ka^{-1}$) and an increase between 3.7 to 3.5 ka BP (140 to 160 $cm*ka^{-1}$) (Fig.2).





Grain size analysis shows an increase in fine sediments, clay (10 %) and silt (70 %), between 5.8 to 3.7 ka BP, with sand decreasing from 100 % to 20 %. Between 3.7 to 0 ka BP the opposite trend occurred, with an increase in sand grains (~84 %) and a decrease of clay (~1 %) and silt (~15 %).

An increase in the Fe/Ca ratio was observed (9 to 15) between 5.8 to 3.7 ka BP with a posterior decrease (15 to 4) toward the top. The iron and calcium alone showed an opposite trend with a decrease between 6 to 3.7 ka BP, an increase between 3.7 to 3 ka BP and posterior decrease toward the top (Fig.3).

Carbonate content showed an increase from 10 % to 50 % between 5.8 to 3.7ka BP. In the interval among 3.7 to 3 ka BP a decrease of the carbonate occurred (from 50 % to 20 %) with a posterior increase toward the top (~80 %) (Fig.4B).

The C/N ratio ranged from 7 to 23 showing a variation between allochthonous and autochthonous organic material. Between 5.8 to 3.7 ka BP the mean value was around 13. Between 3.7 and 3.2 occurred an increase in the values (~18) with a posterior decrease toward the present (Fig.4C).

The total organic carbon ranged from 0.1 to 2 %, with an increase in the values between 4.7 to 3.7 ka BP (Fig.4D). The $\delta^{15}N$ and $\delta^{13}C$ showed the same trend increasing toward the present. The $\delta^{15}N$ ranged from 5 to 15 ‰ and $\delta^{13}C$ range from -40 to -12 ‰ (Fig.4E, F).

## 4 Discussion

### 4.1 Sedimentary processes

Lagoa Salgada paleohydrodynamics show two distinct stages during the mid-late Holocene. The first stage comprised the period between 5.8 to 3.7 ka BP (marine stage) and the second stage from 3.7 to 1.5 ka BP (lagoonal stage).

The marine stage (5.8 to 3.7 ka BP) was characterized by a predominance of fine sediments (Fig.3) and a gradual increase of the sediment deposition toward 3.7 ka BP (Fig.2). According to Castro *et al.* (2014, 2018) and Suguio et al. (1985), the maximum Holocene transgression occurred at ~5 ka BP when the sea level reached ~3m above the modern (Fig.5), causing the submergence of the coastal area. However, the evolution of Paraíba do Sul river delta on the coastal plain formed the Lagoa Salgada initially as an intralagoonal system in a drowned coast around 3,900 years BP (Martin and Suguio, 1992).

In the first stage, the wet condition is dominant in almost all period, with a punctual change from 4.2 to 3.7 ka BP. The gradual increase of the wet condition fed the river increasing fine river discharge and organic material deposition (Fig.3 and 4), indicating changes in the climate condition. The climate changes are inferred by the modification of vegetation type entering the system, and in the source of material deposited (Fig.5D e F). The S-15 core shows that between 5.8 and 3.7 ka BP the Lagoa Salgada was submerged, within a drowning estuary and river flow stagnation occurred with fine sediment decantation in the environment during this period of high sea level (Fig. 3).

Low values of $\delta^{13}C$ (~-25 ‰) and high C/N ratio from organic material (greater than 10) (Meyers, 1997), register an elemental contribution from cellulosic land plants (C3) to the total organic matter input preserved in the sediments, which



were less susceptible to degradation. The $\delta^{13}$C and C/N ratio of the core S-15 show the dominance of C3 plants between 5.8 and 4.8 ka BP with mixed sources between 4.8 and 3.7 ka BP, when a small increase in $\delta^{15}$N occurred, indicating a contribution of another source, as phytoplankton (~ -19 ‰) and C3 plants (~ -25 ‰) (Fig.4C, F), within a period of humid condition. The $\delta^{15}$N shows the source and quality of the organic matter and the influence of terrestrial organic material

during 5.8 to 4.8 ka BP (Fig.4E). This influence is observed by low $\delta^{15}$N values near to 0‰ (Schulz and Zabel, 1999), in which part of the nitrogen demand could come from atmospheric fixation. The increase in the humidity was also seen in cores collected from Lagoa Santa and Lagoa dos Olhos (Table 1) in southeast Brazil, during 7 to 4 ka BP, which favored the vegetation changes with the predominance of C3 plants (Ledru et al., 1998).

Although there was an increase in iron in the sediments, indicating high terrestrial input, the Fe/Ca ratio (Fig. 3C, D)

show an opposite trend. This difference occurred due to the highest amount of calcium deposited in the sediment floor compared to the iron input, which regulates the changes in the Fe/Ca ratio in this environment. The iron input also promoted an increase in the primary productivity and, consequently, the increase in calcium carbonate during the first stage (Fig.3C and 4B).

The second stage comprised the period between 3.7 and 1.5 ka BP. The formation of the sandy barrier caused by sea

transgression favored the creation of lagoonal systems in the delta. During this stage coarse sediments predominate (sand) (~84 %). The Fe/Ca ratio was low with a considerable increase in calcium percentage (~80 %) (Fig.3). According to Castro et al. (2014) a rapid marine regression occurred between 5.5-4.5 ka BP. In the S-15 core, marine regression is identified after 3.7 ka BP when the lake was formed, allowing the input of coarse sediments by erosion with the retreat of the ocean. Lemos (1995) indicated the ages of lake formation at about 2,000 and between 3,090 -3,900 years B.P respectively. The

approximate ages were estimated from different strata of the stromatolites at the edge of the lake.

Geomorphological characteristics and seasonal variability modified the geochemistry of the lake, influencing the sedimentation and precipitation of salts and carbonates that formed biosedimentary structures of stromatolites, thrombolites and oncoids (Silva e Silva et al., 2005, 2008).

Carbonate content show an increase during the second stage as a result of the increasing biological productivity in the

lake, while the C/N ratio shows mixing between C3 plants and phytoplankton as the organic source, with decreasing values. $\delta^{15}$N and $\delta^{13}$C also indicate different sources of organic matter. In this stage, $\delta^{15}$N and $\delta^{13}$C increase towards the top of the core, characterizing changes in vegetation with dominance of C4 plants. C4 plant signature at the top of the core (around 2 to 1.5 ka BP), evidenced by increasing $\delta^{13}$C (~ -10 ‰), was also observed by França et al. (2016) and Ledru et al. (1998) in cores collected in lakes from Southeast Brazil, showing a predominance of herbaceous vegetation during this dryer period.

The climate condition at this time could be influenced by the upwelling system (Laslandes et al., 2006; Nagai et al., 2009, 2016), which favors increasing ocean-land temperature gradient typical of semi-arid climates, corroborating the dominance of C4 plants.

The abrupt change in proxies values in the second stage of the lake show that local climate and the proliferation of microbial communities have modified the geochemistry of the lake and its sedimentation. High $\delta^{15}$N values also suggest

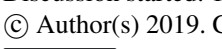



metabolism related to the development of the microorganisms, which gave rise to the estromatolites present in the Lagoa Salgada. The presence of gastropods caused bioturbation in the sediments, affecting the microbial processes and altering the physico-chemical properties of the sediment, by favoring the entry of $O_2$ at the water-sediment interface and N fixation stimulating denitrification (Laverock et al., 2011).

Some species of cyanobacteria have the ability to live in the mud of hypersaline environments and they are halophilic, alkaline (Dupraz et al., 2009) and precipitate carbonates (Xu et al., 2006). Silva *et al.* (2013) identified twenty-one species of cyanobacteria in stromatolites of the Lagoa Salgada, with the most representative being *Microcoleus chthonoplastes* and *Lyngbya aestuarii*, which are diazotrophic cyanobacteria present in coastal microbial mats. In hypersaline lakes, such as Salgada, microbial mats precipitate $CaCO_3$ as a by-product of $CO_2$ capture through photosynthesis by cyanobacteria (Jonkers et al., 2003; Ludwig et al., 2005). The precipitation of $CaCO_3$ that generated the lithification of the microbial mats in the lake are caused by cyanobacteria that increase the pH through photosynthesis in a $CaCO_3$ supersaturated system (Decho and Kawaguchi, 2003).

Radiocarbon dating by Coimbra *et al.* (2000) in the stromatolite head of the Salgada, show the growth of these structures to have begun around 2.2 ± 80 ka BP and finished around 290 ± 80 years B.P. They noticed differences in growth rates of stromatolite relating to the organization of the structure, being better structured at the middle of the head than at the top of the structure, with an average growth rate of 0.05mm/year. In the case of the Salgada, changes in the environmental dynamics and the development of microbial communities after isolation of the marine influence, shown by changes in vegetation type (C4 plants) and an increase in $CaCO_3$ values (80 %), influence the appearance of the stromatolites at around 2.8 ± 8 ka cal.B.P.

## 4.2 The 4.2 event

During the transgressive stage (5.8 to 3.7 ka BP) differences in climate conditions are observed in Southeast Brazil (Fig.6).

Geochemistry data show an increase in productivity between 5 and 4.2ka BP with increasing carbonate and organic carbon percentages (Fig. 6d). The enrichment of organic carbon in the sediment floor is also related with increasing deposition of fine sediments (Fig. 6e), which have the ability to adsorb electrolytes and organic material (Busch and Keller, 1981; Cruz et al., 2013, 2018), thus changing the composition of the sediments.

The wet condition of the environment during this period (5-4.2 ka BP) was characterized by high carbon accumulation and predominance in C3 plants (Fig. 5B, F). High humidity during this period is also characterized by decreasing Mg/Ca ratios in speleothems collected in the Botuvera Cave, Southeast Brazil (Bernal et al., 2016) (Fig. 6G). In that study, Bernal et al. (2016) suggest that most of the changes in rainfall patterns during the Holocene have been driven by the intensity of the South Atlantic Monson Summer (SAMS). SAMS intensification, influenced by the South Atlantic Convergence Zone (SACZ), protrudes as a lower troposphere convective rain belt from the western Amazon to south-eastern Brazil and the South Atlantic (Gandu and Silva Dias, 1998). The precipitation response also results from an adjustment of the Intertropical

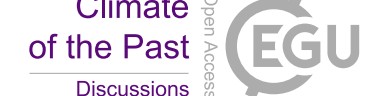

Convergence Zone (ITCZ), which displaces itself according to cooling in Northern Hemisphere and changes in the interhemispheric sea surface temperature (SST) (Cvijanovic et al., 2013). The anomalous southward displacement of the ITCZ shown by dry conditions in the Cariaco Basin (Hughen et al., 1996) (Fig. 6a) indicates increased wet conditions during the transition from the middle to the early Holocene in the south hemisphere.

Higher sand fraction and lower carbonate and iron content reveal a significant change in the environmental condition during the interval of 4.2–3.8 ka BP which could be a regional manifestation of the 4.2ka event in Southeast Brazil. Dry conditions could affect the local vegetation (the mixture of sources shown in Fig. 4F), which leads to the reduction of dense vegetation (C3 plants), increasing erosion and consequently the accumulation of coarser materials. The drought would be caused by a reduction of the intensity of the SAMS and the possible northward displacement of the ITCZ, shown by

increasing wet conditions in Cariaco Basin (Fig. 6A) and drier conditions for the Botuvera cave in southeast Brazil (Fig.6G). In addition, the northward migration of the ITCZ could also have caused the weakening of the upwelling system in southeast Brazil. Upwelling during this period became limited to the subsurface with warm conditions on the surface waters, shown by increasing Mg/Ca ratio in planktonic foraminifera (*G. ruber*) (Fig. 6F) (Lessa et al., 2016).

Several other paleo-archives recovered around Asia (Kaniewski et al., 2018; Kathayat et al., 2018), Europe (Zanchetta et

al., 2016) and Africa (Gasse, 2000) show this drier event between 4.2 and 3.8 ka BP. Arz et al. (2006) suggested that the environmental changes around 4.2 ka BP is an expression of a major drought event, which strongly affected Middle East agricultural civilizations. Sediment core recovered in the Gulf of Oman showed a rather abrupt signature, with climate changes around 4.2 ka BP (Cullen et al., 2000) and a prominent spike of $CaCO_3$ and dolomite indicating the aridity (Fig.6B, C) during the same dry period shown by S-15 core indicating it may correspond to a global event.

The increase in the sand fraction in core S-15 also can be explained by an erosional phase that changed local hydrodynamics, leading to an increase in the coarse deposition as the consequence of a regression of the sea level (Fig. 5B) (Martin and Suguio, 1992). This regression also allows the deposition of the terrestrial organic material, shown by an increase in C/N ratio (Fig. 4C) and a decrease in $\delta^{15}N$ and $\delta^{13}C$ (Fig. 4E, F), causing a decrease of the carbonate accumulation (Fig. 4B). Between 3.9 and 3.7ka BP, a return to the same environmental condition before the event occurred,

with increasing humidity. This period may also have been marked by a new marine transgression, which prevented terrestrial deposition in the study area (Martin and Suguio, 1992).

**5 Conclusions**

The paleohydrodynamics of Lagoa Salgada show a clear adjustment with the variation in the sea level. During the period of the sea level transgression (5.8 to 3.7 ka BP), Lagoa Salgada was submerged, promoting the drowning of a river

and the stagnation of coarse sediment contribution, thus increasing decantation of fine sediment and organic material deposition. This period was also characterized by the dominance of C3 plants and an increase in the sedimentation rate, indicating wetter conditions.



During the transgressive stage (5.8 to 3.7 ka BP), a significant change in climate conditions occurred resulting in a period of aridification, from 4.2 to 3.7 ka BP. The period between 4.2 to 3.7 ka BP was also characterized by changes in the local vegetation, with a reduction of C3 plants and the accumulation of coarse sediments due to increasing erosion. The drought would be caused by a reduction of the intensity of the SAMS due to the northward displacement of the ITCZ.

The regression of the sea level (3.7 ka BP to present) promoted the evolution of Paraíba do Sul river delta on the coastal plain and the formation of the lake system. The lake was formed allowing the input of coarse sediments by erosion with the retreat of the ocean. Abrupt modification in the vegetation type and in the sedimentary deposits was observed in this period, with dominance of C4 plants and decrease in the sedimentation rate indicating a predominance of dry condition on the environment. With the closure of the Lagoa Salgada by the sandy ridges of the delta, geochemical modifications generated internally in the lake allowed the appearance of microbial carpets and stromatolites after 2.8 ka BP.

## Acknowledgements

We thank the support of Project "Stratigraphic, Sedimentological and Geochemical Characterization of Lagoas Salgada, Vermelha and Brejo do Espinho", (PETROBRAS-CENPES). APSC thanks CNPq Process n° 153418/2016-8. This study was financed in part by the Coordenação de Aperfeiçoamento de Pessoal de Nível Superior - Brasil (CAPES) - Finance Code 001".

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



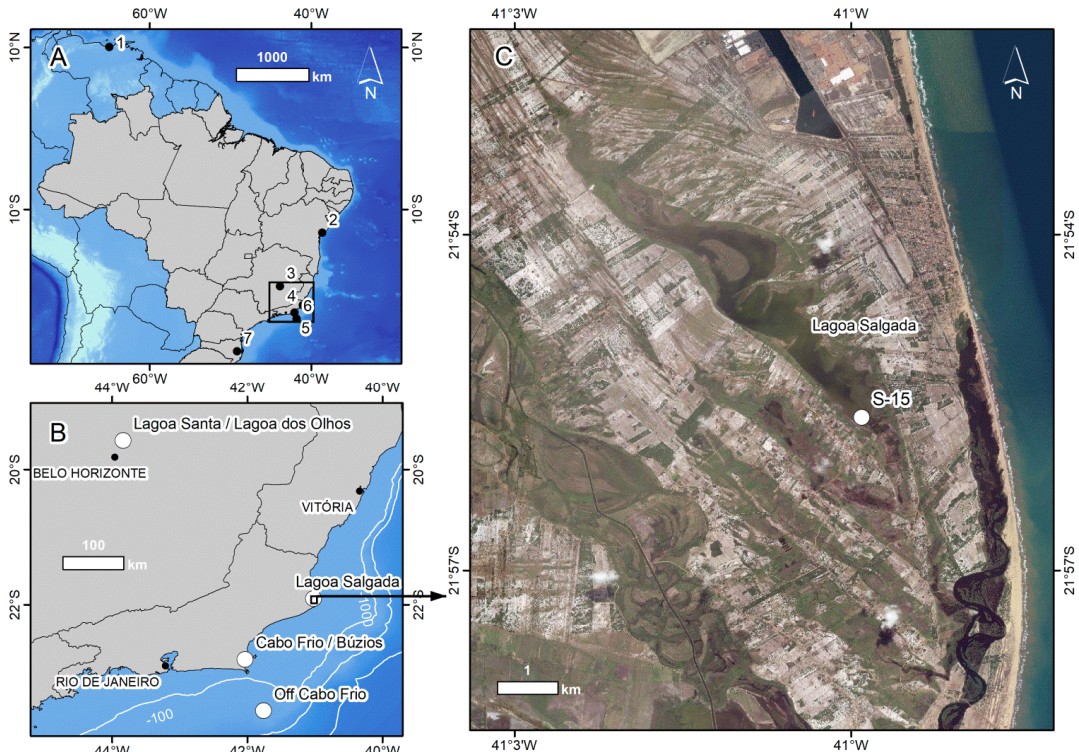

Fig.1: Location map of study area. A) Brazil within South America. Black box indicates Location of B. Numbers refer to sites mentioned in Table 1. B) Southeast Brazil, with state capitals and sites mentioned in the text. C) Location of Core S-15 within Lagoa Salgada. Digital Globe image used as background. Note the seasonal low lake level. Image acquired May 31st, 2017.



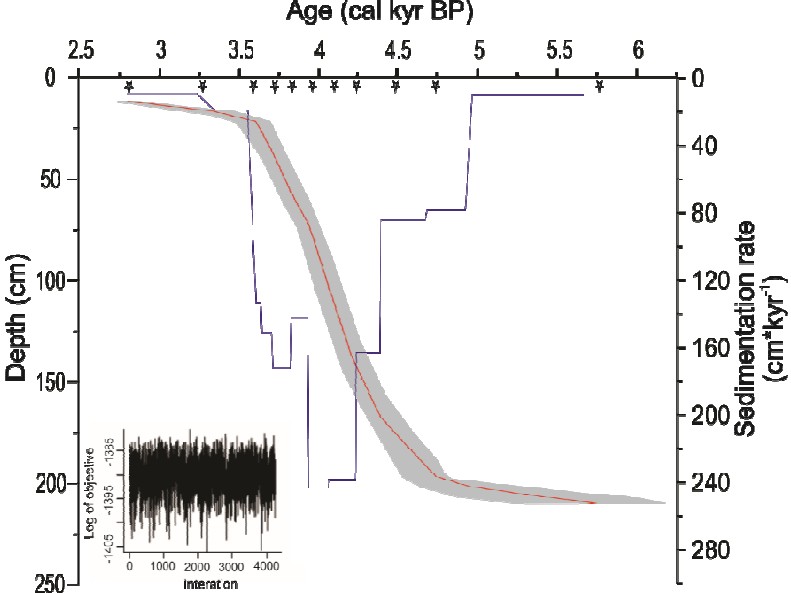

Fig.2: Bayesian age-depth model performed with Bacon (Blaauw & Christeny, 2011) for the core S-15 (red line) and uncertainty (smooth gray curve) from Lagoa Salgada, Rio de Janeiro state, Brazil with sedimentation rate (cm*kyr⁻¹) (blue line). Black stars indicate the position of the 11 radiocarbon dates measured. The bottom left panel shows the interation history.



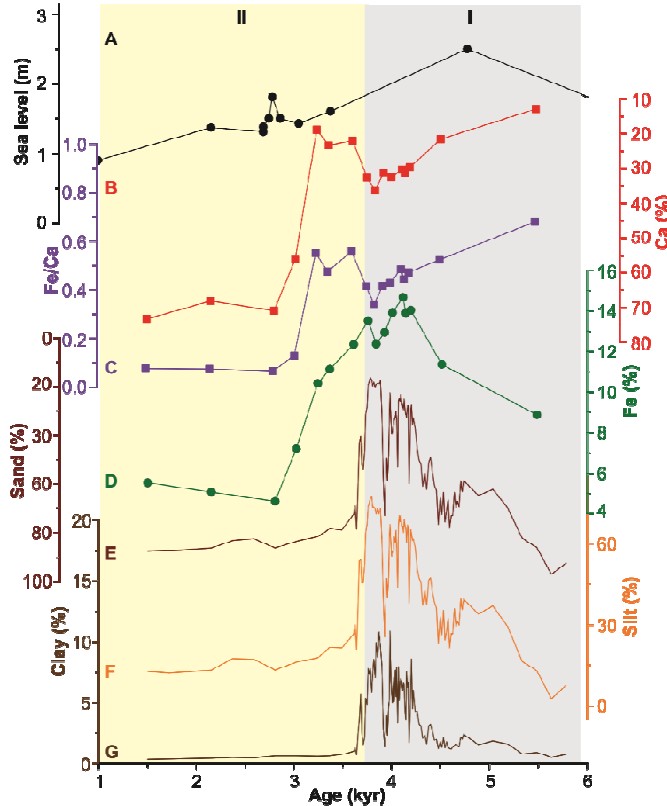

Fig.3: Comparison between sea level changes with sedimentologic records over the past 6 ka BP. (A) Sea level (m) (Castro *et al.*, 2014); (B) Ca (%), (C) Fe/Ca, (D) Fe (%), (E) Sand (%), (F) Silt (%) and (G) Clay (%). The gray and yellow bar indicates two different stages in the last 6ka BP, Marine (I) and Lacustrine (II) stages, respectively.



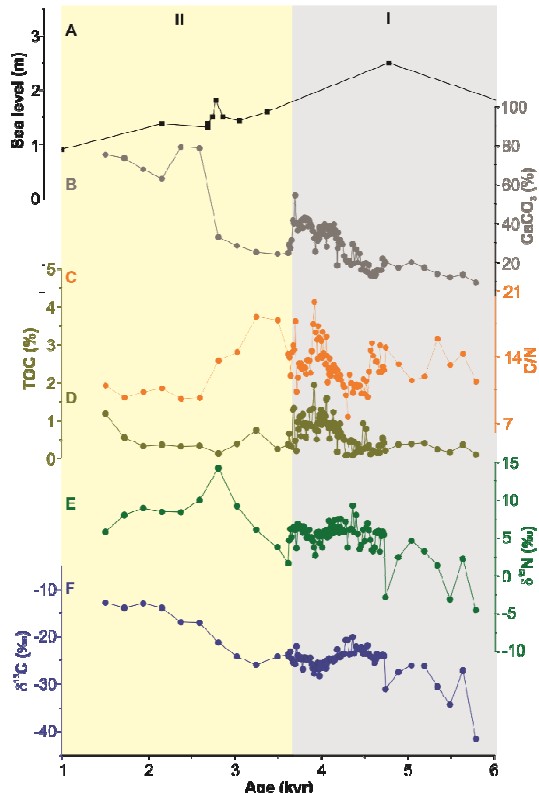

Fig.4: Comparison between sea level changes with geochemical records over the past 6 ka BP. (A) Sea level (m) (Castro *et al.*, 2014); (B) CaCO3 (%), (C) C/N (D) TOC (%), (E) $\delta^{15}N$ (‰) and (F) $\delta^{13}C$ (‰).The gray and yellow bar indicates two different stages in the last 6ka BP, Marine (I) and Lacustrine (II) stages, respectively.





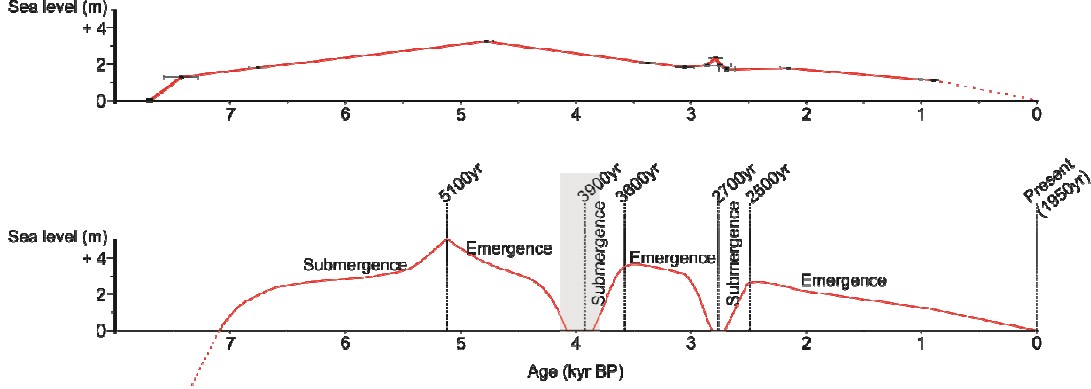

Fig.5: Relative sea-level variation curve for (A) the coast of the Rio de Janeiro state, Brazil (A) (Castro *et al.*, 2014) and (B) Salvador, Bahia state, Brazil (Martin & Suguio, 1992).





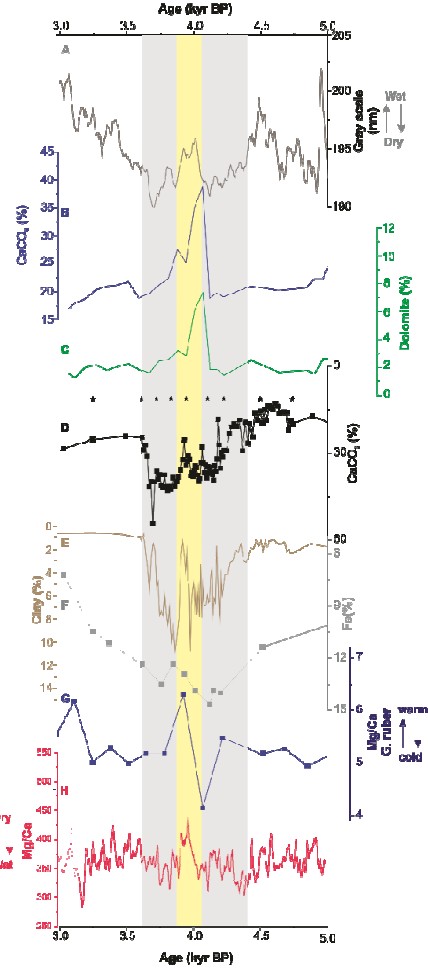

Fig.6: Geochemical records from Lagoa Salgada in comparison with other climate records. A) Gray scale (nm) from Cariaco Basin, Venezuela (Hughen *et al.*, 1996); B) Carbonate content (CaCO₃) (%) and C) Dolomite (%) from Gulf of Oman (Cullen *et al.*, 2000); D) Carbonate content (CaCO₃) (%), E) Clay (%) and F) Fe (%) from this study; G) Mg/Ca

5   *Globigerinoides ruber* from southwest Brazilian coast (Lessa *et al.*, 2016) and H) Mg/Ca Speleothem from Botuvera cave, Southwest Brazil (Bernal *et al.*, 2016). The nine black stars indicate the position of the radiocarbon dates measured in this study. The gray bar emphasizes the period between 4.3 to 3.6 ka BP (wet condition) and the yellow bar shows the 4.2 ka BP event (dry condition).





Table 1: Locations of published records cited.

| N° | Region | Latitude | Longitude | Reference |
|----|--------|----------|-----------|-----------|
| 1 | Cariaco Basin, Venezuela | 10.40 | -65.00 | Hughen *et al.*, 1996 |
| 2 | Salvador, BA, Brazil | -12.58 | -38.58 | Suguio *et al.*, 1985 |
| 3 | Lagoa Santa, MG, Brazil | -21.54 | -41.00 | Ledru *et al.*, 1998 |
| 4 | Lagoa Salgada, RJ, Brazil | -22.44 | -42.01 | ***This study*** |
| 5 | Rio de Janeiro, RJ, Brazil | -23.20 | -41.74 | Castro *et al.*, 2014 |
| 6 | Cabo Frio / Búzios, RJ, Brazil | -27.22 | -49.16 | Lessa *et al.*, 2016 |
| * | Gulf of Oman | -24.39 | -59.04 | Cullen *et al.*, 2000 |
| 7 | Botuvera Cave, SC, Brazil | -27.22 | -49.16 | Bernal *et al.*, 2016 |

*Gulf of Oman is not presented on the map.

Table 2: Ages $^{14}$C obtained from the dating of bulk organic matter to the Core S15.

| LabID | Depth (cm) | $^{14}$C AMS ages | Error (yr BP) | Age (cal yr BP) | Age (2σ cal kyr BP) |
|-------|-----------|-------------------|---------------|-----------------|---------------------|
| BETA363844 | 12 | 2750 | 30 | 2810 | 2.81 |
| BETA363846 | 16 | 3140 | 30 | 3244 | 3.24 |
| AA101944 | 22 | 3582 | 44 | 3613 | 3.61 |
| AA101946 | 38 | 3576 | 43 | 3723 | 3.72 |
| AA101947 | 56 | 3601 | 44 | 3829 | 3.83 |
| AA101948 | 72 | 3832 | 44 | 3942 | 3.94 |
| AA101951 | 110 | 3824 | 44 | 4099 | 4.10 |
| AA101953 | 138 | 3817 | 44 | 4220 | 4.22 |
| AA101955 | 166 | 3952 | 44 | 4496 | 4.50 |
| AA101956 | 196 | 4034 | 44 | 4735 | 4.74 |
| AA101957 | 210 | 5940 | 110 | 5779 | 5.78 |