# Peer review of "Mid-Late Holocene event registered in organo-siliciclastic-sediments of Lagoa Salgada carbonate system, Southeast Brazil"

_Climate of the Past, 2019_

## Referee Comment (RC1) · Anonymous Referee #1 · 20 Apr 2019

General comments

The manuscript by Soares Cruz et alii presents new interesting data on environmental changes, influenced by sea level fluctuations, occurred during the middle and late Holocene in a coastal area of Southern Brazil. The aims of the study are to evaluate both the depositional processes related to sea level changes during the marine and lacustrine stage of Lagoa Salgada site and the Holocene climatic changes occurred in this area during the last 5.8 ka BP.

The work is well structured and easy to read. The methodological approach based on the analysis of lithological features, organic matter geochemistry and $\delta$13C and

$\delta$15N composition performed on a sedimentary core collected from Lagoa Salgada is sound and useful to achieve the aims. The authors show care in the presentation of the results, although some figures need an accurate revision. The discussion presents valid interpretations, but these are not always supported by references to the up-to-date literature dealing with global paleoclimate processes, especially those related to the 4.2 ka event.

My main criticism concerns the interpretation of the stable isotope composition in terms of vegetation dynamics influenced by climate. The climate-vegetation-isotope relationship is hard to understand since this study lacks any local proxy of past vegetation changes.

Overall, the research is well managed, and it has the potential to contribute to the growing literature on the regional specificity of the Holocene climate variability and environmental processes in South America. Before the acceptance, I would recommend an accurate revision of the manuscript according to the following suggestions.

Specific comments

The authors interpret the fluctuations in the $\delta$13C and $\delta$15N composition as the main consequence of change in both vegetation structures and C3/C4 photosynthetic organisms due to climate dynamics (e.g. page 2, lines 9-12; page 4, lines 27-28; page 7, lines 5-9). This is substantially based on the assumption that terrestrial plants are dominated by two distinct vegetation groups employing different photosynthetic pathways (C3 and C4 plants) that determine different $\delta$13C; generally, C3 plants grow under humid conditions while C4 plants under relatively arid conditions. Furthermore, a combination of stable carbon and nitrogen and the C/N ratio of organic matter is used to discriminate the different source of organic matter. The vegetation changes discussed in the manuscript, however, are quite difficult to understand due to the lack of adequate direct proxies (e.g. pollen, plant macro-remains, phytoliths).

In the transitional environments represented by coastal wetlands several factors different from climate changes, such as salinity, light level into the water, and impact of human activities, among others, can shape the communities of primary producers, influencing the stable isotopic composition of the organic matter into the sediments. Sea level fluctuations also produce major geomorphic and ecological changes in coastal areas, which have the potential to modify the sedimentological processes and affects the communities of primary producers, even without a direct influence of climate. For example, input of saltwater into a coastal wetland may determine the development of communities of halophilic plants and aquatic algae, featured by photosynthetic species that may present a wide range of $\delta$13C and $\delta$15N values (see Duarte et al. 2018, Frontiers in Marine Sciences, doi: 10.3389/fmars.2018.00298). The formation of coastal dunes may trigger the development of plant communities dominated by C4 species of Poaceae in a context not influenced by climate changes. In the hydrosere, often occurring in coastal wetlands without a direct relationship with climate, there is the succession of different environmental stages characterized by peculiar sedimentological processes and photosynthetic organisms (C3 and C4 species), which have the potential to produce major changes in the stable isotopes composition of the organic matter. The authors do not seem to adequately consider the high variety of ecological situations that can influence the isotopic composition in coastal sites. Therefore, I would like to suggest them to comment in the text on the possible uncertainties of the applied methodology in the study of coastal sites and to discuss occurrence / exclusion of other possible factors that influence the composition of $\delta$13C and $\delta$15N and the C/N ratio in their study area.

Paragraph 4.2 must be integrated by references to recent research dealing with the 4.2 ka event. I would suggest the authors to look for recent literature focussed on this climate change characterized by a high regional specificity. To this purpose there is a special issue of Climate of the Past devoted to 4.2 ka event with contributions from various regions of the world.

Figures 2-6 of the pdf version of the manuscript I downloaded shows a low quality in

terms of image resolution. Besides, the Figures 3 and 6 show curves too close to each other with overlapped scales that limit the readability of the data. I would suggest the authors to check this graphic material and improve its quality.

Technical corrections

Page (P) 2, Line (L) 28: check the correct version of the Bacon program

P 4, L 7: include space between '3.7' and 'ka'

P 4, L 10: add 'ka' after '3.2'

P 4, L 21: include space between '3' and 'm'

P 6, L 1: change 'estromatolites' to 'stromatolites'

P 6, L 14: write '2200±80 BP'

P 6, L 16: include space between '0.05' and 'mm/year'

P 6, L 14: write '2800±8 BP'

P 6, L 23: include space between '4.2' and 'ka'

P 6, L 31: change 'Monson' to 'Monsoon'

P 7, L 6: include space between '4.2' and 'ka'

P 7, L 24: include space between '3.7' and 'ka'

---

## Referee Comment (RC2) · Anonymous Referee #2 · 21 Apr 2019

I have now finished the review of the manuscript "Mid-Late Holocene event registered in organo-siliciclastic-sediments of Lagoa Salgada carbonate system, Southeast Brazil by Soares Cruz et al., The topic of the manuscript fits well with the aims of the journal and I do think it deserves publication on Climate of the Past. The text is generally easy to read even if some improvement in the quality of the figures is required.

However, revisions are required before publication. In particular, more methodological details are required. In my opinion grain size analysis and geochemistry need to be supported by bio-stratigraphic analysis in order to reconstruct the paleao-environments in a transitional area. The transition from marine to brackish faunal assemblages were

extensively used for these kinds of analysis. The authors should better explain why they did not use these proxies. Furthermore, there is an extensive discussion on the vegetation dynamics but no clear investigations on past vegetation changes is reported in the manuscript. This makes the discussion not always well supported by the data. Finally, the references are not always up to date, with some relevant recent papers focused on the 4.2 ka event missing from the introduction/discussion.

So, I suggest moderate revisions before the publication of this paper on Climate of the past.

I detailed below the points that should be improved before the publication.

Introduction

The introduction needs some work. This is now not well focused. It is unclear for the reader why the Lagoa Salgada is important for this kind of investigation. You deeply described the global importance of the 4.2 ka event but it is presently unclear why your case study is important to investigate this. Furthermore, the cited references on the 4.2 events are not always up to date. So an effort in the improvement of this section is strongly required.

Methods Methodology is generally wll described but some additional data are required. Can you please provide more details about the coring operation? Did you use a vibracore or a hand corer? What is the elevation of the top of the core with respect to the current msl? Furthermore, how did you reconstruct the depositional environment? It seems that you did not use meio or macrofaunal assemblages to define the palaeo-environments. This is a bit surprising because these proxies are widely used to this purpose. Grain sizes usually should be corroborated by these data. So, you should at least explain why you did not perform this kind of analysis (maybe lack of faunal assemblages??).

Discussions I generally agree with the discussions but I don't think they are always

based on the results. As I said before, I think the palaeo-environmental reconstructions are a bit weak because only based on grain size and geo-chemical analysis. Furthermore, there is a large discussion focused on the vegetation but no pollen (or similar) analysis are reported. However, my major concern is related to the use of Martin & Suguio, 1992 RSL reconstruction provided in figure 5. This sequence of high and low sea-level stands needs to be better explained. From an isostatic point of view this is quite complex to justify. Do you think there are other factors controlling the sea-level evolution in this area? This is a major part of your discussion and it is now not fully explained in the manuscript. I do understand the RSL highstand reported by Castro et al., 2014 at about 5000 BP. On the contrary, the yo-yo shape of the RSL curve reported by Martin & Suguio, 1992 needs clarification.

---

## Author Comment (AC1) · 20 May 2019

We thank Reviewer #1 for the very constructive review of our manuscript. Below, we provide a point-by-point response to the Reviewer #1. To facilitate the review, we copied the Reviewers′ comments in black and inserted our comments after that. Reviewer #1(Specific comments): The authors interpret the fluctuations in the 13C and 15N composition as the main consequence of change in both vegetation structures and C3/C4 photosynthetic organisms due to climate dynamics (e.g. page 2, lines 9-12; page 4, lines 27-28; page 7, lines 5-9). This is substantially based on the assumption that terrestrial plants are dominated by two distinct vegetation groups employing different

photosynthetic pathways (C3 and C4 plants) that determine different 13C; generally, C3 plants grow under humid conditions while C4 plants under relatively arid conditions. Furthermore, a combination of stable carbon and nitrogen and the C/N ratio of organic matter is used to discriminate the different source of organic matter. The vegetation changes discussed in the manuscript, however, are quite difficult to understand due to the lack of adequate direct proxies (e.g. pollen, plant macro-remains, phytoliths). In the transitional environments represented by coastal wetlands several factors different from climate changes, such as salinity, light level into the water, and impact of human activities, among others, can shape the communities of primary producers, influencing the stable isotopic composition of the organic matter into the sediments. Sea level fluctuations also produce major geomorphic and ecological changes in coastal areas, which have the potential to modify the sedimentological processes and affects the communities of primary producers, even without a direct influence of climate. For example, input of saltwater into a coastal wetland may determine the development of communities of halophilic plants and aquatic algae, featured by photosynthetic species that may present a wide range of 13C and 15N values (see Duarte et al. 2018, Frontiers in Marine Sciences, doi: 10.3389/fmars.2018.00298). The formation of coastal dunes may trigger the development of plant communities dominated by C4 species of Poaceae in a context not influenced by climate changes. In the hydrosphere, often occurring in coastal wetlands without a direct relationship with climate, there is the succession of different environmental stages characterized by peculiar sedimentological processes and photosynthetic organisms (C3 and C4 species), which have the potential to produce major changes in the stable isotopes composition of the organic matter. The authors do not seem to adequately consider the high variety of ecological situations that can influence the isotopic composition in coastal sites. Therefore, I would like to suggest them to comment in the text on the possible uncertainties of the applied methodology in the study of coastal sites and to discuss occurrence / exclusion of other possible factors that influence the composition of 13C and 15N and the C/N ratio in their study area. Authors: We agree that several factors can change the

environmental condition of the coastal areas. Thus, we added on page 2, in the introduction, other factors that can change the environment beyond the climate. We also mention in the discussion part, page 5 (line 20-24), the problems with the nitrogen and carbon isotopes in the discrimination of the primary producers and we emphasize the importance of the pollen analysis to discuss the vegetation changes. Unfortunately, we don't have pollen analysis made in this core or in this lagoon. However, we use pollen analysis made in other lagoons of southeast Brazil to corroborate with ours proxy and emphasize the influence of the climate in this region. In page 6 (line 18-24), we reinforce the idea of the other factors, as the input of saltwater into the coastal wetland and the formation of coastal dunes, which can trigger the development of plant communities as a result of the competitive advantages of salt-tolerant species, but we also showed that in pollen data analyzed from cores collected from lagoons in southeast Brazil, without influence of the coastal dynamics, also show changes in vegetation as a result of the climate alterations through the Holocene, making the climate an important environmental modifying factor in this region. Reviewer #1: Paragraph 4.2 must be integrated by references to recent research dealing with the 4.2 ka event. I would suggest the authors to look for recent literature focused on this climate change characterized by a high regional specificity. To this purpose there is a special issue of Climate of the Past devoted to 4.2 ka event with contributions from various regions of the world. Authors: Agree. Done. Reviewer #1: Figures 2-6 of the pdf version of the manuscript I downloaded shows a low quality in terms of image resolution. Besides, the Figures 3 and 6 show curves too close to each other with overlapped scales that limit the readability of the data. I would suggest the authors to check this graphic material and improve its quality. Authors: Agree. Done. We have substituted for a better resolution. Reviewer #1: Technical corrections Page (P) 2, Line (L) 28: check the correct version of the Bacon program P 4, L 7: include space between '3.7' and 'ka' P 4, L 10: add 'ka' after '3.2' P 4, L 21: include space between '3' and 'm' P 6, L 1: change 'estromatolites' to 'stromatolites' P 6, L 14: write '2200_80 BP' P 6, L 16: include space between '0.05' and 'mm/year' P 6, L 14: write '2800_8 BP' P 6, L 23: include space between '4.2'

and 'ka' P 6, L 31: change 'Monson' to 'Monsoon' P 7, L 6: include space between '4.2' and 'ka' P 7, L 24: include space between '3.7' and 'ka' Authors: All the technical corrections were done.

---

## Author Comment (AC2) · 20 May 2019

We thank Reviewer #2 for the very constructive review of our manuscript. Below, we provide a point-by-point response to the Reviewer #2. To facilitate the review, we copied the Reviewers' comments and inserted our comments after that.

Reviewer #2 (Specific comments): The introduction needs some work. This is now not well focused. It is unclear for the reader why the Lagoa Salgada is important for this kind of investigation. You deeply described the global importance of the 4.2 ka event but it is presently unclear why your case study is important to investigate this. Furthermore, the cited references on the 4.2 events are not always up to date. So an

effort in the improvement of this section is strongly required.

Authors: Agree. Done. We dedicated one paragraph trying to explain the importance of the Lagoa Salgada in this investigation.

Reviewer #2: Methods Methodology is generally well described but some additional data are required. Can you please provide more details about the coring operation? Did you use a vibracore or a hand corer? What is the elevation of the top of the core with respect to the current msl? Furthermore, how did you reconstruct the depositional environment? It seems that you did not use meio or macrofaunal assemblages to define the palaeoenvironments. This is a bit surprising because these proxies are widely used to this purpose. Grain sizes usually should be corroborated by these data. So, you should at least explain why you did not perform this kind of analysis (maybe lack of faunal assemblages??).

Authors: The core was collected using a Vibracore (page 3, line 3) with a PVC tube. The core was split in two halves and sliced every 2cm for resolution. The core head is located at present day sea level, estimated from the best available Digital Elevation Model, built using the most detailed topographical map available at: ftp://geoftp.ibge.gov.br/cartas_e_mapas/bases_cartograficas_continuas/bc25/rj/versao2018/ (IBGE, 2018). Detailed information is available at Nota Tecnica (nota-tecnica_bc25_rj_2018-05-23.pdf). DEM generation method and detailed information is available at the Metadata document, (Metadados_MDE_RJ25.pdf). All documents are available at the IBGE site. Vertical accuracy is ~1m, with the DEM classified every meter. Present water level at Salgada Lake is at 0m msl for both the vectorized topographical map and DEM. One year monthly observation of the water level at the lake, yields an average of a few centimeters of depth, with no water being the most common (personal communication from Douglas Rosa da Silva; Kátia Leite Mansur; Leonardo Fonseca Borghi de Almeida, authors of Distribution and Growth Morphology of the Recent Microbialites: the Case of Lagoa Salgada, Rio de Janeiro – Brazil). Thus, the Salgada Lake core top is considered to be at present-day sea level.

We didn't use faunal assemblages (e.g. foraminifera) due to the low preservation potential at this age. We only observed Quinqueloculina sp. and Ammonia sp., which bring no additional faunal information showing just an evaporitic environment with extreme faunal restriction.

Reviewer #2: Discussions I generally agree with the discussions, but I don't think they are always based on the results. As I said before, I think the palaeo-environmental reconstructions are a bit weak because only based on grain size and geo-chemical analysis. Furthermore, there is a large discussion focused on the vegetation but no pollen (or similar) analysis are reported. However, my major concern is related to the use of Martin & Suguio, 1992 RSL reconstruction provided in figure 5. This sequence of high and low sea-level stands needs to be better explained. From an isostatic point of view this is quite complex to justify. Do you think there are other factors controlling the sea-level evolution in this area? This is a major part of your discussion and it is now not fully explained in the manuscript. I do understand the RSL highstand reported by Castro et al., 2014 at about 5000 BP. On the contrary, the yo-yo shape of the RSL curve reported by Martin & Suguio, 1992 needs clarification.

Authors: We agree that the lack of pollen data leaves our discussion a little weak. However, we use pollen analysis made in lakes in southeast region to corroborate with ours proxies and emphasize the influence of the climate in this region.

We just used the Martin and Suguio (1992) sea level curve, in the last paragraph of the discussion, to demonstrate that the sea level change during 4.2 kaBP may also have caused a change in the environmental conditions of the region. However, we included in the discussion, that the lack of data during this period, in both sea level curves (Castro et al 2014 and Martin and Suguio, 1992), make this hypothesis of sea level regression merely speculative, and the influence of climate change a more plausible alternative to the environmental changes that occurred during this period.

Please also note the supplement to this comment:

https://www.clim-past-discuss.net/cp-2019-27/cp-2019-27-AC2-supplement.pdf

[Figure]

**Supplement:**

Altitude of core head was estimated from the most detailed topographical map available at [ftp://geoftp.ibge.gov.br/cartas_e_mapas/bases_cartograficas_continuas/bc25/rj/versao2018/](ftp://geoftp.ibge.gov.br/cartas_e_mapas/bases_cartograficas_continuas/bc25/rj/versao2018/) (IBGE, 2018). Detailed information is available at *Nota Tecnica* in annex (nota-tecnica_bc25_rj_2018-05-23.pdf). DEM generation method and detailed information is available at the Metadata document, in annex (Metadados_MDE_RJ25.pdf). Vertical accuracy is ~1m, with the DEM classified every meter.

Present water level at Salgada Lake is at 0m msl for both the vectorized topo map and DEM. One year monthly observation of the water level at the lake, yields an average of a few centimeters of depth, with no water being the most common (personal communication from Douglas Rosa da Silva; Kátia Leite Mansur & Leonardo Fonseca Borghi de Almeida, authors of *Distribution and Growth Morphology of the Recent Microbialites: the Case of Lagoa Salgada, Rio de Janeiro – Brazil*, in annex).

Thus, the Salgada Lake core top is considered to be at present-day sea level.

[Figure]

Fig 1: Screen capture of chosen DEM for the area, available at [https://downloads.ibge.gov.br/downloads_geociencias.htm#](https://downloads.ibge.gov.br/downloads_geociencias.htm#)

[Figure]

Fig 2: IBGE DEM draped over Sentinel-2 imagery. White line is 0m msl (above). Core site: red star. The excavation for Porto do Açu is observed in the north of the satellite imagery (below).

---

## Author Response (AR2)

June 26, 2019.

Dear Alesio Roveri,

We have accepted the requested suggestions and explained within the methodology the reason for not using foraminifera assemblages in this work. We remain in doubt about the figure suggested for the modification, since we understand that Figure 4 is in high resolution and in the same style as Figure 3. Figure 5 is the one that really needed to be improved; and in this way we have inverted the ages in the x axis according to the remaining figures and removed the text of figure 5B, to make it clearer. The resolution of both Figures 4 and 5 have been improved and we hope we have met the demands of this prestigious journal.

Sincerely yours

Catia F Barbosa